# Bacterial Colonization as a Possible Source of Overactive Bladder Symptoms in Pediatric Patients: A Literature Review

**DOI:** 10.3390/jcm10081645

**Published:** 2021-04-13

**Authors:** Katarzyna Kilis-Pstrusinska, Artur Rogowski, Przemysław Bienkowski

**Affiliations:** 1Department of Pediatric Nephrology, Wroclaw Medical University, Borowska 213, 50-556 Wroclaw, Poland; 2Faculty of Medicine, Cardinal Stefan Wyszyński University in Warsaw, Collegium Medicum, Kazimierza Wóycickiego 1/3, 01-938 Warsaw, Poland; arogowski@op.pl; 3Department of Obstetrics and Gynecology, Mother and Child Institute, 01-211 Warsaw, Poland; 4Department of Psychiatry, Medical University of Warsaw, Nowowiejska 27, 00-665 Warsaw, Poland; pbienko@yahoo.com

**Keywords:** overactive bladder, urinary microbiome, children, adolescents

## Abstract

Overactive Bladder (OAB) is a common condition that is known to have a significant impact on daily activities and quality of life. The pathophysiology of OAB is not completely understood. One of the new hypothetical causative factors of OAB is dysbiosis of an individual urinary microbiome. The major aim of the present review was to identify data supporting the role of bacterial colonization in overactive bladder symptoms in children and adolescents. The second aim of our study was to identify the major gaps in current knowledge and possible areas for future clinical research. There is a growing body of evidence indicating some relationship between qualitative and quantitative characteristics of individual urinary microbiome and OAB symptoms in adult patients. There are no papers directly addressing this issue in children or adolescents. After a detailed analysis of papers relating urinary microbiome to OAB, the authors propose a set of future preclinical and clinical studies which could help to validate the concept in the pediatric population.

## 1. Introduction

Overactive Bladder (OAB) is a common condition that is known to have a significant impact on daily activities and quality of life [1,2,3,4]. In the pediatric population, OAB may not only burden child development but also have a negative impact on the family situation [5,6]. OAB is a form of lower urinary tract dysfunction caused by involuntary detrusor contractions during the filling phase. It is characterized by urgency and increased voiding frequency with or without incontinence in the absence of urinary tract infection [7]. OAB is not associated with neurological and anatomical alterations of the lower urinary tract. The recent International Children’s Continence Society (ICCS) document proposes using the term daytime lower urinary tract (LUT) conditions to group all functional bladder problems in children [8]. OAB is the second most common bladder dysfunction following nocturnal enuresis [9]. Comorbid conditions associated with bladder dysfunction include urinary tract infections, vesicoureteral reflux, constipation, and encopresis [1,9,10,11]. In addition, there appears to be an increased frequency of behavioral and neurodevelopmental issues [5,6]. One should bear in mind that the course and clinical picture of OAB in children differ substantially from that observed in adult patients. On a theoretical ground, there are several unique factors to consider in the context of pediatric OAB, including the role of general child development, mode of delivery, mother microbiome, maturation of the nervous and endocrine systems, and development of the urinary tract [12,13,14,15,16].

The overall OAB prevalence in the pediatric population ranges from 1.5% to 36.4% [17,18,19]. It is known that the peak incidence occurs between 5 and 7 years of age, with a higher prevalence in males [1,12,20]. Overactive bladder decreases with age [13,21]. Results of a cross-sectional survey of 19,240 Korean schoolchildren identified the overall incidence of OAB, defined as urgency with or without incontinence, was 17%. The highest incidence of 23% was noted in five-year-old children and the lowest (12%) in 13-year-old patients [17]. Although OAB can resolve itself spontaneously, urinary symptoms can persist into adulthood [14,22]. The prevalence estimates for urge UI ranged from 1.8% to 30.5% in the European population, from 1.7% to 36.4% in the US population, and from 1.5% to 15.2% in the Asian population [2].

The pathophysiology of OAB is not completely understood and is believed to be multifactorial [15,23,24,25,26]. One proposed hypothesis is that dysbiosis of an individual urinary microbiome could induce or potentiate OAB symptoms [16,27,28,29,30]. The major aim of the present review was to identify data supporting the role of bacterial colonization in overactive bladder symptoms in children and adolescents. The second aim of our study was to identify major gaps in current knowledge and possible areas for future clinical research.

## 2. Methods

The present work was not intended to directly answer a well-defined clinically meaningful question or to modify current practice and thus could hardly fulfill the definition of a systematic review. The paucity of clinical data and lack of randomized clinical trials on the role of the microbiome in overactive bladder symptoms in children precluded a meaningful scoping review approach [31]. Hence, the present review was designed as a narrative mini-review with an assumption that an increase in the number of full-text papers may allow us to use a scoping or systematic review approach in the future [32].

Full-text papers focused on the association between the microbiome and overactive bladder symptoms were identified by searching MEDLINE and Google Scholar from inception to December 2020. Keywords included the following terms: “bacterial colonization” OR “microbiome” OR “urobiome” OR “dysbiosis” AND “overactive bladder” OR “urgency” OR “urge incontinence”. A manual search of reference lists of relevant papers was also performed. One hundred and fifty-six papers were identified and screened for eligibility. Twelve original studies on microbiome and OAB were finally included in the review process. All efforts were made to identify studies on the pediatric population through manual search of abstracts and method sections for the age of recruited subjects.

Three independent reviewers screened abstracts for eligibility. Clinical studies on human subjects written in English with an available full text were included. Book chapters and conference abstracts were checked but excluded if not followed by a full-text publication.

## 3. Theoretical and Clinical Background

### 3.1. OAB

The pathophysiology of OAB is not completely understood and is believed to be multifactorial [24,25,29,33]. One theory is that the urgency and related symptoms stem from a cortical immaturity of the centers responsible for controlling urination [15,34]. Voluntary and coordinated urination is developed over time. In the first year of life, voiding is mainly controlled by the brainstem [35]. Then, cortical inhibitory pathways and the pontine micturition center along with periaqueductal grey matter, anterior cingulate gyrus, and the autonomic, somatic, and sensorial autonomic nervous systems are developed, and urination becomes voluntary [36]. The prefrontal cortex starts to maintain top-down control over more primitive afferent pathways of the brain, such as the limbic and paralimbic systems [24].

Another concept, the so-called “bladder-brain dialogue”, suggests a mutual interaction between the brain and the bladder rather than unidirectional control by the brain [26,33,36]. The role of inflammation in OAB is also under investigation [25,37]. Bladder biopsies from patients with OAB without urinary tract infection (UTI) showed inflammatory changes [38]. Ghoniem et al. demonstrated upregulation of a selective subset of proinflammatory cytokines and chemokines in patients with OAB [39]. Urgency is attributed to abnormal neuromuscular signaling resulting from the stimulation of cholinergic receptors, which causes involuntary bladder muscle contractions [40,41]. However, in urodynamic studies, detrusor overactivity was only observed in approximately 58% of women with urge UI [42]. Therapy with anticholinergic drugs which mainly inhibit the function of efferent neurons in the detrusor muscle is ineffective for approximately half of patients that use them [43]. Comparable observations have been made concerning children. In case series of children with voiding dysfunction symptoms, detrusor overactivity detected by urodynamic testing was present in 52% to 58% of patients compared with 5% to 18% of asymptomatic children [44,45]. Moreover, a prospective multicenter study reported a poor correlation between symptoms and urodynamic testing in children with incontinence [46]. In a study by Bael et al., 60 of 91 children with urgency did not have evidence of an overactive bladder during bladder filling when urodynamic testing. In addition, there was a poor correlation between urodynamic findings and the response to the treatment [46].

An active exploration of etiological factors and pathophysiological mechanisms standing behind OAB symptoms in children is a prerequisite for developing safe and efficacious treatment strategies.

### 3.2. Microbiome

The term microbiome refers to the bacterial milieu present within various environment niches. Each individual’s microbiome is unique and adapts during life as a result of environmental and genetic influences [47,48]. The maternal microbiome is a dominant factor in the development of the neonatal skin, oral mucosa, and nasopharyngeal microbiome, regardless of the delivery mode [49,50]. In the next period of life, the maternal microbiome further affects the child’s microbiome through multiple transmission routes. What is interesting is a metatranscriptomic analysis of bacterial strains specific to mother-infant pairs suggests that gastrointestinal bacteria were not only transferred from the maternal gut to the infant gut environment but that the bacteria adapted effectively to the infant gut [50].

The microbiome can assist in maintaining healthy states in the human body including homeostasis and immune defense [48]. On the other hand, alterations to the microbial community structure may be also implicated in disease [51,52].

### 3.3. Urinary Microbiome

In the past, the urinary tract (without urethra) was considered to be sterile under normal conditions. The advances in bacterial assessment in the past decade, particularly 16S rRNA gene sequencing and expanded quantitative urine culture (EQUC), have shown that the urinary tract is not sterile [27,28,38,52,53,54,55]. Up to 80% of bacteria can be isolated using modified culture techniques for a sample that has been classified as having no growth according to the standard method [54]. According to an analysis of the literature performed by Morand et al., the urinary tract bacterial microbiome contains 21.4% of the known prokaryotic diversity associated with human beings (464 species in common), and it shares 23.6% of species with the human gut microbiota (350 species in common, 62.3% of the urine species) [56]. Females have predominantly *Lactobacillus* and *Gardnerella* species, while males carry *Corynebacterium*, *Staphylococcus*, and *Streptococcus* as dominant species [38,55]. In addition, females tend to have a more heterogeneous urinary microbiome. The species found in urine can be pathogenic or commensal. At least 60.0% of the urine microbiota is not reported in the literature as causing human UTI [56].

The characteristics and role of urinary microbiota are currently debated [33,51,52]. Urinary tract microbiota influences UTI [16,57,58]. Modification of bacterial components in urine has been associated with kidney stones, bladder cancer, and urinary incontinence [16,27,51,59]. In the context of the urinary microbiome, the gut and vaginal microbiome are also of interest, since there is evidence that implies that there is some degree of cross-talk between the bacterial flora of these organs [60,61,62,63].

The majority of urinary microbiome studies focus on adult subjects and papers concerning children are scarce. The pediatric studies on the urinary microbiome are collected in Table 1. The natural history of the urinary microbiome remains mostly unexplored. However, to understand the role of the urinary microbiome in disease states, both in children and adults, it is necessary to know what may constitute a healthy urinary microbiome and how it develops in early childhood.

The relation between the urinary microbiome of parents and their children is largely unknown because to date no study has directly addressed this topic. Given the anatomic relationship between the vagina and the urinary tract, the vaginal microbiome may be relevant to the urinary microbiome. However, Hickey et al. stated that the vaginal microbiome of adolescent girls was not compatible with that of their mothers [60]. This suggests, as was the case with the gut microbiome, that the acquisition of the female “adult-form” microbiome is more of a maturation of the microbiome rather than a transition in the species representation. Currently, the data concerning the vaginal microbiome of girls are not consistent. According to a review paper by Smith and Ravel, the prepubertal vaginal microbiome is dominated by a variety of anaerobes, diphtheroid, coagulase-negative *Staphylococci*, and *E. coli*, while the postmenarcheal vaginal microbiome is most similar to adult vaginal microbiomes, dominated by *Lactobacillus* [61]. However, a prospective longitudinal study of perimenarcheal girls documented that *Lactobacillus* dominated the vaginal microbiome before the onset of menarche [60]. In addition, *Gardnerella vaginalis*, classically considered to be pathogenic, was found in up to one-third of perimenarcheal subjects. Probably, these bacteria play a commensal role in the prepubertal period.

The vaginal microbiome can be of importance in an analysis of a healthy urinary microbiome and its dysbiosis or perturbations. Urine, intestinal and vaginal microbiomes are interconnected. For example, intestinal bacteria may colonize the vaginal entrance and periurethra, and then ascend up the urethra to the bladder. The vaginal microbiome may be a natural line of defense against invading uropathogens or it may alter the urinary microbiome [62].

Kinneman et al. [57] assessed the urinary microbiome in children younger than 48 months undergoing a urinary catheterization, with and without UTI. A urinary microbiome was identified in every child. The 5 most abundant families were tissierellaceae, prevotellaeae, veillonellaceae, enterobacteriaceae, and comamonadaceae. The 5 most abundant genera were *Prevotella*, *Peptoniphilus*, *Escherichia*, *Veillonella*, and *Finegoldia*. Alpha diversity, which refers to the number of different species in a single site, did not differ by age, gender, antibiotic use 15 days to three months before the urine sample was obtained, maternal ethnicity, country of origin, delivery mode, or probiotic use. Decreased diversity and changes in the compositions of urinary microbiome were observed in children with standard culture-positive UTI. The authors noticed that antibiotic use affected the urinary microbiome only for a short time (up to two weeks).

Kassiri et al. [66] examined the urinary microbiome in 20 prepubertal males (aged 3 months-8 years; median age 15 months) with and without prior antibiotic exposure. The majority of patients had representation from *Staphylococcus* and *Varibaculum* species and to a lesser extent *Peptoniphilus* and *Actinobaculum*. Several of the detected genera have been previously identified in the urine of adult men. However, urinary microbial communities profiled in children were different from those described in adults. For example, *Staphylococcus* and *Corynebacterium* were present in children but were not dominant. The authors stated that the composition of the urinary microbiome in children may begin to develop early in life and evolve over time, becoming more stable in adulthood. Moreover, the study also showed differences in both the urinary and gastrointestinal microbiome in children with prior antibiotic exposure, confirming the effect of drugs on the child microbiome.

Forster et al. [65] performed a cross-sectional analysis of the urine microbiome of children with neuropathic bladders. *Enterobacteriaceae* are the most predominant bacteria in the urine microbiomes, along with *Staphylococcus*, *Streptococcus*, and *Enterococcus*. There was no difference in the urine microbiome between children with UTI, asymptomatic bacteriuria, and negative standard cultures. It has been observed that the route of catheterization may affect the composition of the urine microbiome. Children who catheterize their urethra have a higher proportion of *Staphylococcus*, while the urinary microbiome of patients who catheterize through a Mitrofanoff was composed of *Enterobacteriaceae* family bacteria.

In summary, the urinary microbiome in the pediatric population has just begun to be explored. Further studies that focus on the potential variables influencing the urinary microbiome are needed.

### 3.4. Review of Studies on Urinary Microbiome and OAB

The first studies investigating the urinary microbiome in patients with urgency concerned women suffering from urge UI who had no signs of infection [53,54]. Research studies confirmed urinary bacterial DNA and the relation of bladder polymicrobial community to certain clinical variables such as baseline urgency urinary incontinence episodes, treatment response, and post-treatment UTI risk [29]. When comparing women with and without urge UI, *Gardnerella* and *Lactobacillus gasseri* were associated with urge UI, while *Lactobacillus crispatus* was detected most frequently in controls, indicating the possibility of a protective effect of *Lactobacillus crispatus* in preventing the development of urge UI [27]. Karstens et al. revealed that the urine microbiome composition of women with normal bladder function and women with urge UI not only varies in the type of bacteria that are present but also in the number of different bacteria and abundance of these bacteria [69]. Women with more severe urge UI symptoms have decreased microbial diversity in their urinary microbiomes. Moreover, the authors noted that of the nine species found to be overrepresented in the urine of patients with urge UI, five bacteria reported as pathogens causing UTI are not routinely detected by routine cultures. This suggests that a persistent low-grade infection by such bacteria could potentially be responsible for the irritating symptoms of urge UI. In another study, it has been established that the uropathogenic bacteria *Proteus* was more commonly isolated from women with OAB. On the other hand, the genus *Lactobacillus* was present less commonly in urine from OAB patients when compared to urine taken from controls [70]. The results are in line with studies describing a significantly greater prevalence of *Lactobacillus* in controls compared to patients with bladder symptoms. The protective role of *Lactobacilli* is explained by their ability to produce bacteriocins, which have activity against uropathogenic bacteria [63].

The above-mentioned studies suggest that perturbations in the urinary microbiome, a state referred to as *dysbiosis*, may predispose to the development of OAB. However, the kind of microbiome diversity connected with OAB is not clear. Thomas-White et al. found that women with urge UI had a different and more diverse microbiome as compared to unaffected women [71]. Among patients treated with solifenacin, an anticholinergic drug, a better response was observed in women with fewer bacteria and a less diverse microbiome whereas non-responders had a community that often included bacteria not typically found in responders. However, in most studies, a decrease in species diversity was associated with urgency UI [69,70].

One should bear in mind that the above studies were cross-sectional in nature and it is not clear whether the differences in the urinary microbiome are the cause or consequences of OAB. It is possible that the urinary frequency typically associated with urgency urinary incontinence alters the microbial community. Significant new findings have highlighted the non-barrier role of the urothelium, especially its sensory functions [24,33,72,73]. As there is clear evidence of communication between the bladder and the brain, it is biologically plausible that the urinary microbiota may play some role in this communication. Therefore, the urothelial sensory signaling and its alteration may be responsible for bladder dysfunction.

## 4. Conclusions and Future Directions for Studies on Urinary Microbiome and Pediatric OAB

The increasing body of evidence tends to indicate some relationship between qualitative and quantitative characteristics of individual urinary microbiome and OAB symptoms in adult patients [29,69,70,71]. Surprisingly little is known about the possible associations between the urinary microbiome and OAB symptoms in the pediatric population. In fact, we were unable to identify papers that would specifically address this issue in children or adolescents.

Given the variety of etiopathological concepts and clinical presentations of OAB and the plethora of its somatic and psychological consequences in pediatric patients, studies on the role of the urinary microbiome in OAB could be of clear theoretical and practical importance. The following paragraphs may provide some basic ideas and impetus for research on the link between the urinary microbiome and OAB in children and adolescents.

From a theoretical point of view, future clinical studies could target specific quantitative and/or qualitative traits of the urinary microbiome as correlates of bladder physiology and pathophysiology assessed with the aid of a urine test, an ultrasound of the urinary tract, a bladder diary, dysfunctional elimination symptom questionnaires, and urodynamic methods. Microbiological and molecular approaches could help to identify bacterial species and their metabolic products directly responsible for local alterations in the urothelial milieu even in the absence of obvious symptoms of lower urinary tract infection.

From a practical point of view, it remains to be established whether qualitative and/or quantitative features of the urinary microbiome could be risk or protective factors for the development of OAB in pediatric patients. As clinicians are typically confronted with sick children rather than healthy at-risk individuals, one may also wish to know whether urinary microbiome fingerprints could provide some markers of OAB symptom severity and long-term prognosis in already diagnosed cases. For obvious reasons, it would be of value to correlate the history of lower urinary tract infections and cumulative antibiotic exposure with alterations in the urinary microbiome and OAB symptomatology.

Last but not least, future randomized clinical trials could address the role of antibiotics and probiotics in the primary or secondary prevention of OAB as well as in the treatment of OAB in children and adolescents. Gender, age, hormonal status as well as neuropsychiatric, metabolic (e.g., obesity), renal, urological, and gynecological comorbidities may pose a set of hypothetical factors modifying possible associations between the urinary microbiome and OAB in the pediatric population.

## Figures and Tables

**Table 1 jcm-10-01645-t001:** Pediatric studies of the urinary microbiome.

	Study Group: Age, *n*	Key Results
Lucas et al. [64]	60 female children divided into four developmental groups: 0–3 m/o (*n* = 15), 4–10 m/o (*n* = 15), 2–6 y/o (*n* = 15), 7–12 y/o premenstrual girls (*n* = 15)	Significant shifts in the perianal and periurethral/perivaginal (PUPV) microbiome compositions during childhood, corresponding to important developmental milestones. Significant differences in the PUPV microbiome of girls with a history of UTI, likely influenced by both the UTI and the antibiotic exposure.
Curley et al. [58]	Review	Review of the literature on the role of the microbiome in recurrent UTIs, focusing on female pediatric patients when able.
Forster at al. [65]	34 children with neuropathic bladders with UTI (*n* = 11, mean age 11 y/o), ASB (*n* = 19, mean age 8.8 y/o), and with negative urine culture (*n* = 4, mean age 15 y/o)	The most predominant bacteria in the urine microbiomes are from the *Enterobacteriaceae* family. No difference in the urine microbiome between children with UTI, ASB, and negative urine cultures. Route of catheterization may affect the composition of the urine microbiome.
Kinneman et al. [57]	85 children < 48 months of age (72 less than 24 months)	Urinary microbiome was identified in every child, even in 3 subjects less than 30 days of age. Changes in microbiome diversity and composition were observed in subjects with a standard culture-positive UTI.
Kassiri et al. [66]	Prepubertal boys (*n* = 20, ages 3 months–8 years; median age 15 months)	The first characterizations of the urinary microbiome in prepubertal males. Defining the baseline healthy microbiome in children may lay the foundation for understanding the long-term impact of factors such as antibiotic use in the development of a healthy microbiome as well as the development of future diseases.
Kispal et al. [67]	12 children, 6–17 years (median age at the time of surgery 11 years)	After bladder augmentation, the native urinary bladder and augmented intestinal segments host similar microbiota despite their distinct differences of originating mucosal anatomy. Age at sampling had a statistically significant influence on β-diversity at the genus level.
Gerber et al. [16]	Review	Review of the literature on the effects of the microbiome on urologic diseases that affect the pediatric patient, including UTI, urge urinary incontinence/overactive bladder, and urolithiasis.
Ollberding et al. [68]	49 cases (delivery < 37 weeks gestation) and 48 controls (delivery ≥ 37 weeks gestation)	No difference in taxa richness, evenness, or community composition between cases and controls or for gestational age modeled as a continuous variable.

UTI-urinary tract infection; ASB-asymptomatic bacteriuria.

## Data Availability

Not applicable.

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
