# Peer review of "Bacterial Colonization as a Possible Source of Overactive Bladder Symptoms in Pediatric Patients: A Literature Review"

_jcm, 2021, doi:10.3390/jcm10081645_

Round 1

Reviewer 1 Report

Although I find it an interesting paper proposing future research for urinary microbiome and OAB in children, the paper would benefit from the following improvements.

- Methods
1) What was the total number of articles searched and how many studies are included in this review?

- Data review
1) Why is research on urinary microbiome so rare in children? Are there any specific technical issues to consider?

2) It would be better to make a table showing pediatric studies of the urinary microbiome.

Author Response

We are grateful to the Reviewer for his/her valuable comments which helped us to improve our manuscript.

 Reviewer: Although I find it an interesting paper proposing future research for urinary microbiome and OAB in children, the paper would benefit from the following improvements.

- Methods
1) What was the total number of articles searched and how many studies are included in this review?

Authors: One hundred and fifty six papers were identified and screened for eligibility. Twelve original studies on microbiome and OAB were finally included in the review process. The information has been added to the new version of the ms. (Methods, page 2).

Reviewer: Data review
1) Why is research on urinary microbiome so rare in children? Are there any specific technical issues to consider?

Authors: The urinary microbiome has just begun to be explored, especially in pediatric population. It is a typical situation in general medicine that first studies on new subject are performed in adult population and next in pediatric groups. Pediatric population is much smaller than adult one and the number of pediatric researches is also lower, which implies fewer studies in the pediatric population and with fewer participants. In addition, it is difficult to obtain homogenous pediatric groups for studies.

Moreover, it is easier to obtain biological material for testing from an adult than a child. There are different ways to collect urine from children for testing. These may affect the results of the urinary microbiome examination.

Each individual's microbiome is unique and adapts during life as a result of environmental and genetic influences. Childhood is a period of intensive development. Nowadays, it is unknown how the urinary microbiome develops in early childhood and what may constitute a healthy microbiome. There are many potential variables that may influence the child urinary microbiome, for example term and mode of delivery, mother microbiome, maturation of the nervous and endocrine systems, and development of the urinary tract. Taking into account all the factors involved in child's development makes urinary microbiome testing difficult.

Reviewer:
2) It would be better to make a table showing pediatric studies of the urinary microbiome.

Authors: The table (Table 1) has been added to the ms. as suggested by the Reviewer. 

Reviewer 2 Report

Authors from well-respected Polish institutions have composed a narrative mini-review on bacterial colonization as a possible source of overactive bladder symptoms in pediatric patients. I accepted this review with interest as I had come across very little of such material in the literature previously. However, that goes hand-in-hand with what I consider to a major concern of this paper… zero studies directly address this topic. Specifically, as the authors themselves note in their discussion, “Surprisingly little is known about the possible associations between urinary microbiome and OAB symptoms in the pediatric population. In fact, we were unable to identify papers which would specifically address this issue in children or adolescents.” I have a difficult time grasping the concept of a review, be it narrative or systematic, that addresses a topic on which no papers have been published.

An increasing amount of literature has been published on OAB + urinary microbiome in adults (which the authors do cite, and has been reviewed extensively in other recent publications), but the authors do not make clear in their abstract/introduction why it is particularly important to study this topic in children/why it would be expected to deviate from adults.

Additional comments:

Abstract

-Remove “all” (“all supporting data”) because this is misleading about the scope of a narrative review

Introduction

-Several sentences missing necessary referencing, including all sentences in first paragraph, last 2 sentences of second paragraph, first sentence of third paragraph

-Overall, introduction can be condensed into 1-2 paragraphs. Many sentences in the second paragraph do not seem to be of direct relevance to the primary aim of this review, and would be more appropriately moved to the discussion or eliminated. Specifically, first 3 paragraphs can be feasibly merged into one paragraph.

- “One of the new hypothetical causative factors of OAB is dysbiosis of an individual urinary microbiome.” Requires source, particularly as this is a major point of justification for the present study.

Methods:

-Why no studies on animal models?

-“bacterial colonization” OR “microbiome” OR “urobiome” OR “dysbiosis” AND “overactive bladder” OR “urgency” OR “urge incontinence”…. Why not search for pediatrics?

Data Review

-First paragraph missing several citations.

Author Response

We are grateful to the Reviewer for his/her valuable comments which helped us to improve our manuscript.

Reviewer: Authors from well-respected Polish institutions have composed a narrative mini-review on bacterial colonization as a possible source of overactive bladder symptoms in pediatric patients. I accepted this review with interest as I had come across very little of such material in the literature previously. However, that goes hand-in-hand with what I consider to a major concern of this paper… zero studies directly address this topic. Specifically, as the authors themselves note in their discussion, “Surprisingly little is known about the possible associations between urinary microbiome and OAB symptoms in the pediatric population. In fact, we were unable to identify papers which would specifically address this issue in children or adolescents.” I have a difficult time grasping the concept of a review, be it narrative or systematic, that addresses a topic on which no papers have been published.

Authors: In our opinion, the idea of narrative review does not exclude the possibility that authors review the literature and find no relevant data directly supporting a given concept. Still, indirect evidence may be reviewed (e.g. evidence from other age groups), hypotheses gathered, and researchers in the field informed of a research gap and attractive avenues for research.

From a more practical perspective, we strongly believe that studies on adults may stimulate reasonable research in pediatric patients. Hence, one may wish to review the field to support (as in our review) or discourage scientifically-oriented pediatricians to cross the border between age groups and disciplines.

However, if the Editors see the title of the ms. misleading, we are prone to change the title towards "a perspective" or "a perspective review".

Reviewer:

An increasing amount of literature has been published on OAB + urinary microbiome in adults (which the authors do cite, and has been reviewed extensively in other recent publications), but the authors do not make clear in their abstract/introduction why it is particularly important to study this topic in children/why it would be expected to deviate from adults.

Authors: Characteristics of OAB  in children is different than in adults. In pediatric population OAB is the second most common bladder function disorder. It seems to be underreported. OAB does not have only a significant  impact on child's daily activities and quality of life, but it also impacts his/her families. The information was given in the Introduction section. For children idiopathic OAB is rather typical, for adults rather secondary. In the pathophysiology of pediatric OAB factors relating to child’s development should be taken into account. Similarly, there are many potential variables that may influence the child urinary microbiome, for example term and mode of delivery, mother microbiome, maturation of the nervous and endocrine system, and development of the urinary tract. Given the above, we assumed that the subject “OAB and urinary microbiome in children” deserves more attention.

In line with the Reviewer's comment, we have added a brief paragraph on the differences between children and adult patients to the Introduction section (page 2, lines 17-20, the new version of the ms.).

Reviewer: Abstract

-Remove “all” (“all supporting data”) because this is misleading about the scope of a narrative review.

Authors: The word “all” has been deleted. The same word has been deleted from the last paragraph of Introduction.

Reviewer: Introduction

-Several sentences missing necessary referencing, including all sentences in first paragraph, last 2 sentences of second paragraph, first sentence of third paragraph.

Authors: We are grateful the Reviewer for this valuable remark. The missing necessary references have been added.

Reviewer:

-Overall, introduction can be condensed into 1-2 paragraphs. Many sentences in the second paragraph do not seem to be of direct relevance to the primary aim of this review, and would be more appropriately moved to the discussion or eliminated. Specifically, first 3 paragraphs can be feasibly merged into one paragraph.

Authors: As suggested, the first three paragraphs have been merged into one paragraph. The next two paragraphs have been combined into one paragraph. Now Introduction consist of two paragraphs. We have left all the sentences of the second paragraph unchanged as they refer to the specificity of OAB in the pediatric population.

Reviewer:

- “One of the new hypothetical causative factors of OAB is dysbiosis of an individual urinary microbiome.” Requires source, particularly as this is a major point of justification for the present study.

Authors: We are grateful the Reviewer for this valuable remark. The relevant references have been added.

Reviewer:

Methods:

-Why no studies on animal models?

Authors: The idea to study or illustrate pediatric disorders with the aid of animal models is interesting but difficult from a practical point of view. First, it seems that animal models of overactive bladder in juvenile subjects do not exist. Second, regardless of subject's age, one can hardly identify any preclinical studies on urobiome and overactive bladder in laboratory animals.  

Reviewer:

-“bacterial colonization” OR “microbiome” OR “urobiome” OR “dysbiosis” AND “overactive bladder” OR “urgency” OR “urge incontinence”…. Why not search for pediatrics?

Authors: All the papers were manually screened for the age of groups recruited to the study. We have made this issue more clear in the new version of the ms. (page 2, the Method section).

Reviewer: Data Review

-First paragraph missing several citations.

Authors: The missing references have been added.

Reviewer 3 Report

Bacterial colonization, as a possible source of overactive bladder (OAB) symptoms in pediatric patients, has been studied in very few instances. The authors are proposing a roadmap for better understanding of the association between urinary microbiome and OAB symptoms in this population, from theoretical and practical points of view, with suggestions for randomized clinical trials.

The Title – I would suggest “Literature review”, rather than “Perspective review”

The introduction is quite lengthy, and if I may make some suggestions, aside from shortening it:

On page 2, eliminate the sentence starting with “Moreover…” on line 49, ending on line 51

On page 2, line 59 sentences starting with “One should…” and the subsequent ones to the end of the paragraph, can be moved to the first paragraph of the Introduction – it does set the stage for the uniqueness of the children, and the reason for the this literature review.

Page 1 – line 38 – a “strikethrough word “disorder” – remove it

Page 1 – line 41 – neurodevelopment should read neurodevelopmental

Page 1 – line 44 – in “…with increasing age” can delete “increasing”

Page 1 – line 45 – replace “showed” with “identified”

Page 2 – line 46 – replace “as” with “to be”

Page 2 – line 55 – sentence starting with “One…” can be replaced with something like “One proposed hypothesis is that dysbiosis of an individual urinary microbiome can be causing OAB”

Methods – line 80 – “All efforts were done…” should read “All efforts were made…”

Page 2 – remove the strike through words “Data review”

Page 3 – lines 100 and 103 – replace “overactive bladder” with abbreviation already defined - “OAB”

Page 3 – line 120-121 – the sentence is a duplicate of the one on page 2 line 55, which I suggested to be rephrased

References – spacing and format should be following the Authors’ Instructions – as the Introduction section will be shorter, ensure some references are deleted and sequence adjusted

Table 1 – format to eliminate unnecessary spacing

Table 1 – “taxa richness” may not be the suitable term for what the authors desire to express – it is a term used more frequently in ecological environment

Author Response

We are grateful to the Reviewer for his/her valuable comments which helped us to improve the manuscript. The changes in the ms. body have been marked with yellow color.

 Reviewer

“The Title – I would suggest “Literature review”, rather than “Perspective review””

Authors:
The Title has been changed as suggested (see the new version of the ms.).

 Reviewer:

"The introduction is quite lengthy, and if I may make some suggestions, aside from shortening it:

On page 2, eliminate the sentence starting with “Moreover…” on line 49, ending on line 51."

Authors: 

The sentence has been removed.

Reviewer:

"On page 2, line 59 sentences starting with “One should…” and the subsequent ones to the end of the paragraph, can be moved to the first paragraph of the Introduction – it does set the stage for the uniqueness of the children, and the reason for the this literature review."

Authors: 

We are grateful to the Reviewer for this remark. The above mentioned sentences have been moved to the first paragraph of the Introduction. The Introduction has been shortened -eliminated sentences has been marked with the yellow color and strikethrough text.

Reviewer:

"Page 1 – line 38 – a “strikethrough word “disorder” – remove it

Page 1 – line 41 – neurodevelopment should read neurodevelopmental

Page 1 – line 44 – in “…with increasing age” can delete “increasing”

Page 1 – line 45 – replace “showed” with “identified”

Page 2 – line 46 – replace “as” with “to be” "

Authors:

 Corrected as suggested.

Reviewer:

"Page 2 – line 55 – sentence starting with “One…” can be replaced with something like “One proposed hypothesis is that dysbiosis of an individual urinary microbiome can be causing OAB”. "

Authors:

The sentence has been changed as suggested by the Reviewer.

Reviewer:

"Methods – line 80 – “All efforts were done…” should read “All efforts were made…”. "

Authors:

 It has been changed as suggested.

Reviewer:

"Page 2 – remove the strike through words “Data review” "

Authors:

Corrected as suggested.

Reviewer:

"Page 3 – lines 100 and 103 – replace “overactive bladder” with abbreviation already defined - “OAB” "

Authors:

It has been changed as suggested.

Reviewer:

"Page 3 – line 120-121 – the sentence is a duplicate of the one on page 2 line 55, which I suggested to be rephrased."

Authors:

The duplicated sentence has been removed.

Reviewer:

"References – spacing and format should be following the Authors’ Instructions – as the Introduction section will be shorter, ensure some references are deleted and sequence adjusted."

Authors:

Corrected as suggested.

Reviewer:

"Table 1 – format to eliminate unnecessary spacing."

Authors:

 Corrected as suggested.

Reviewer:

"Table 1 – “taxa richness” may not be the suitable term for what the authors desire to express – it is a term used more frequently in ecological environment."

Authors:

Thank you for the remark. The term “taxa richness” was given by the authors of the cited paper. We listed it according to the original article.

We are grateful to the Reviewer for his/her valuable comments which helped us to improve the manuscript. The changes in the ms. body have been marked with yellow color.

 Reviewer

“The Title – I would suggest “Literature review”, rather than “Perspective review””

Authors:
The Title has been changed as suggested (see the new version of the ms.).

Reviewer:

"The introduction is quite lengthy, and if I may make some suggestions, aside from shortening it:

On page 2, eliminate the sentence starting with “Moreover…” on line 49, ending on line 51."

Authors: 

The sentence has been removed.

Reviewer:

"On page 2, line 59 sentences starting with “One should…” and the subsequent ones to the end of the paragraph, can be moved to the first paragraph of the Introduction – it does set the stage for the uniqueness of the children, and the reason for the this literature review."

Authors: 

We are grateful to the Reviewer for this remark. The above mentioned sentences have been moved to the first paragraph of the Introduction. The Introduction has been shortened -eliminated sentences has been marked with the yellow color and strikethrough text.

Reviewer:

"Page 1 – line 38 – a “strikethrough word “disorder” – remove it

Page 1 – line 41 – neurodevelopment should read neurodevelopmental

Page 1 – line 44 – in “…with increasing age” can delete “increasing”

Page 1 – line 45 – replace “showed” with “identified”

Page 2 – line 46 – replace “as” with “to be” "

Authors:

 Corrected as suggested.

Reviewer:

"Page 2 – line 55 – sentence starting with “One…” can be replaced with something like “One proposed hypothesis is that dysbiosis of an individual urinary microbiome can be causing OAB”. "

Authors:

The sentence has been changed as suggested by the Reviewer.

Reviewer:

"Methods – line 80 – “All efforts were done…” should read “All efforts were made…”. "

Authors:

 It has been changed as suggested.

Reviewer:

"Page 2 – remove the strike through words “Data review” "

Authors:

Corrected as suggested.

Reviewer:

"Page 3 – lines 100 and 103 – replace “overactive bladder” with abbreviation already defined - “OAB” "

Authors:

It has been changed as suggested.

Reviewer:

"Page 3 – line 120-121 – the sentence is a duplicate of the one on page 2 line 55, which I suggested to be rephrased."

Authors:

The duplicated sentence has been removed.

Reviewer:

"References – spacing and format should be following the Authors’ Instructions – as the Introduction section will be shorter, ensure some references are deleted and sequence adjusted."

Authors:

Corrected as suggested.

Reviewer:

"Table 1 – format to eliminate unnecessary spacing."

Authors:

 Corrected as suggested.

Reviewer:

"Table 1 – “taxa richness” may not be the suitable term for what the authors desire to express – it is a term used more frequently in ecological environment."

Authors:

Thank you for the remark. The term “taxa richness” was given by the authors of the cited paper. We listed it according to the original article.

Round 2

Reviewer 1 Report

The authors have revised their manuscript profoundly. 

Reviewer 2 Report

Thank you for the opportunity to review a revised version of the manuscript. The authors did elaborate somewhat on the potential importance of differentiating between pediatric and adult OAB, and also added additional citations as requested. Unfortunately, this manuscript remains a review, "narrative," “perspective,” or otherwise, based on a topic on which no original papers have been published.

Author Response

Reviewer: "Thank you for the opportunity to review a revised version of the manuscript. The authors did elaborate somewhat on the potential importance of differentiating between pediatric and adult OAB, and also added additional citations as requested. Unfortunately, this manuscript remains a review, "narrative," “perspective,” or otherwise, based on a topic on which no original papers have been published."
Authors:
We tried to make this point clear in the first and second version of the ms. We did the data review with all efforts and standards required for a narrative review, clearly indicated the paucity of data, showed the roadmap for future studies, analyzed potential difficulties but also clinical significance of improved understanding of 'pediatric' OAB.

We understand and accept that the Reviewer may have different expectancies when reading the ms. We appreciate all the comments and suggestions expressed in the original Reviewer's report.

Reviewer 3 Report

Thank the authors for accommodating the reviewers' recommendations for this literature review